# Could Training in an Anatomical Model Be Useful to Teach Different Neovagina Surgical Techniques? A Descriptive Study about Knowledge and Experience of Techniques for Neovagina Surgery

**DOI:** 10.3390/jcm9113722

**Published:** 2020-11-19

**Authors:** María Luísa Sanchez-Ferrer, Grigoris Grimbizis, Michele Nisolle, Enrique Salmeron-González, Luis Gómez-Pérez, Francisco Sánchez del Campo, Maribel Acién

**Affiliations:** 1Department of Obstetrics & Gynecology, “Virgen de la Arrixaca” University Clinical Hospital, 30120 El Palmar (Murcia), Spain; 2Institute for Biomedical Research of Murcia, IMIB-Arrixaca, 30120 El Palmar (Murcia), Spain; 3First Department of Obstetrics and Gynecology, Aristotle University of Thessaloniki, 54636 Thessaloniki, Greece; grigoris.grimbizis@gmail.com; 4Department of Obstetrics and Gynecology, University of Liège, CHR Liège, 4000 Liège, Belgium; michelle.nisolle@chuliege.be; 5Plastic Surgery and Burns Service, University and Polytechnic Hospital La Fe, 46026 Valencia, Spain; enrikes900@gmail.com; 6Department of Urology, “San Juan” University Clinical Hospital, University Miguel Hernández, 03550 Alicante, Spain; luisgope@gmail.com; 7Department of Histology and Anatomy, University Miguel Hernández, 03550 Alicante, Spain; sanchez.campo@umh.es; 8Department of Obstetrics & Gynecology, “San Juan” University Clinical Hospital, University Miguel Hernández, 03550 Alicante, Spain; macien@umh.es

**Keywords:** thiel anatomical model, neovagina, surgical skills

## Abstract

Neovagina surgery in patients with vaginal agenesis is rare. No consensus exists regarding the best surgical technique. The aims of the current study were to show a new Thiel-embalmed cadaveric model to teach the surgical steps for different techniques of neovagina surgery and to evaluate opinions of this surgical teaching procedure. Four techniques—modified McIndoe, Vecchietti, Davydov, and vulvoperineal pediculated flaps—were recorded using an external camera and/or laparoscopic vision during their execution in a dissection room on “feminized” male cadavers. To determine the opinion of this teaching model, we designed an anonymous online survey that was available to participants via a computer application. After watching the video, more than 92% of participants agreed that feminized male cadavers were an excellent procedure for teaching these surgical techniques. Before watching this video, the most employed techniques were the McIndoe and Vecchietti procedures. After watching the video, modified McIndoe and vulvoperineal flaps were preferred by participants because they were considered to be easier to perform. It was considered that this model was useful for training neovagina techniques and, moreover, it should be recommended before techniques were performed on a real patient. Further investigation is needed to validate this model.

## 1. Introduction

Neovagina surgery for patients with vaginal agenesis is rare. The prevalence of vaginal agenesis due to Rokitansky Syndrome is 1 in 5000 (range of 1 per 4000 to 10,000 females) [1]. An additional pathology responsible for vaginal agenesis and the need for neovagina surgery is androgen insensitivity syndrome, whose incidence in females is 1:20,000 [2]. In addition, with the recent increase in sex change surgery, greater interest in neovagina surgery exists. The Frank technique (primary vaginal dilation) offers satisfactory results in 69–94% of cases [3]. Surgery is an option for women who have been unsuccessful with dilators or who prefer surgery after a thorough discussion of the advantages and disadvantages of the different techniques. If this fails, or if the patient refuses self-manipulation, surgery is indicated, even though this may be the first choice when there are other associated malformations that require intervention. Compared with primary vaginal dilation, vaginoplasty complications are much more common and include, depending on the technique used, bladder or rectal perforation, graft necrosis, hair-bearing vaginal skin, fistulae, diversion colitis, inflammatory bowel disease, and adenocarcinoma [3]. In addition, the literature describes different techniques of vaginal and laparoscopic approaches (modified McIndoe, Vecchietti, Popoff, Davydov, pediculated flaps etc.). Each of these is effective and none are superior to the others. These techniques are based on the dissection of a new space in the rectovesical septum, performed either vaginally or laparoscopic assisted (Appendix A). This relatively infrequent surgery should be performed in a few specialized centers [4]. However, the reality is that these procedures are not regulated. Therefore, acquiring experience in this type of surgery is a challenge. When dealing with a relatively low number of operations of a special type, the chosen technique should be simple, safe, and effective [5]. Furthermore, rapid surgical innovation in minimally invasive procedures, devices, and surgical techniques have complicated the learning landscape. Fortunately, surgical simulation has evolved to fill the educational void. Whether it is through skill generalization or skill transfer, surgical simulation has shifted learning from the operating room back to the classroom. Educational simulation programs are necessary to improve specialist knowledge and skill, and to facilitate competence in this kind of surgery. After carrying out a bibliographic search, we did not find reports of models to train these surgical techniques before performing them on patients. The ideal model would withstand repetitive use and would not be prohibitively expensive. Computer-based teaching models and low-fidelity silicone replicas demonstrate the location of anatomic structures and their relationships in a three-dimensional space [6]. High-fidelity laparoscopic trainers are often expensive. Animal models have proven effective in simulating several surgical techniques, such as robotic hysterectomies, but are expensive and anatomical differences may limit their usefulness [7]. Use of human cadavers is a simulation aid that allows for surgical practice via extremely detailed and life-like reproductions of each structure and decreased ethical risk associated with the use of animal surgical laboratories [8].

The objective of this work was to show a new Thiel-embalmed “feminized” male cadaver model for teaching the surgical steps of four different techniques of neovagina surgery to treat vaginal agenesis cases and to evaluate opinions of this surgical learning procedure in comparison with other learning models.

## 2. Experimental Section

The procedure was performed on Thiel-embalmed cadavers, which allowed the vaginal approach and abdominal cavity pneumo-insufflation to, more precisely, reproduce the surgical technique, by both vaginal and laparoscopic approaches. The procedure was carried out in the dissection room at the School of Medicine of the Miguel Hernández University in San Juan, Alicante, Spain. Previously, we “feminized” male cadavers. This involved removing the penis and testicles, and reconstructing the labia using the skin of the penis and scrotum. The space between the base of the scrotum and the anus was exposed to perform the different “neovagina” procedures. This preparation is a novel approach for the hands-on training of neovagina surgery on cadaveric models and allows a very real dissection of the spaces.

We made recordings of four different surgical procedures to perform neovagina surgery (modified McIndoe, modified Vecchietti, Davydov, and vulvoperineal pediculated flaps) (Appendix A) with an external camera for vaginal procedures, and laparoscopic vision during the execution of the abdominal approach, allowing the visualization of anatomical elements. A final video [9] was produced that showed the four neovagina surgical techniques.

To determine opinions regarding this teaching model, we designed an anonymous online survey (Appendix B). To design the survey, the literature was searched to identify the surgical training models that had been evaluated for neovagina techniques. However, none were found. Thus, we designed a survey (Appendix B) to explore the opinions of specialist doctors and attendants and speakers at the recording session regarding their experiences with these surgical training models. We conducted a short pilot test with five students to examine the comprehension of the questions. Then, the survey was sent to the remaining participants. During the presentation of the survey, we explained that we used “feminized” Thiel-embalmed male cadavers to perform neovagina surgery. We clarified that this embalmed method allows the cadaver to be moved to perform vaginal surgery and laparoscopic procedures in a manner similar to that used with patients. This survey was made available to program participants via a computer application containing a link to the video [9] and the survey itself [10]. The survey was sent to four groups: (1) Attendants and speakers at the European Society of Human Reproduction and Embryology (ESHRE) Campus Symposium “Gynecological pathologies at adolescence,” organized at Miguel Hernandez University, San Juan, Alicante, Spain. These participants were chosen because they watched this hands-on session live when it was broadcast to the audience, (2) students and professors of the “Master of pelvic floor dysfunctions” (PFM) of Miguel Hernandez University, Elche, Alicante, Spain. This University Master’s course on multidisciplinary pelvic floor surgery provides a professional update on pelvic floor diseases and their management. The Master’s course includes an update on the anatomy, pathophysiology, and clinical workup of the patient with pelvic floor dysfunctions, including urogenital anomalies. In particular, the course provides hands-on teaching for the management of these patients and for the multidisciplinary instruments in the dissection room using Thiel-embalmed cadavers, (3) gynecologists affiliated to the Valencian gynecological Society (SOGCV), and (4) the Murcian Gynecological Society (SGM). These participants were chosen to explore the opinion of the general gynecologist about the method, whereas the other groups were considered to have some experience with Thiel-embalmed cadavers and with neovagina procedures in the scope of their practice.

## 3. Results

### 3.1. Characteristics of the Participants

#### 3.1.1. Specialty

The response rate was 25.9%. A total of 133 surveys were returned (Table 1). A substantial majority of survey participants were gynecologists. In the ESHRE group, 12/13 (92.3%) were gynecologists. Only 1/13 (7.7%) had another specialty different from gynecology, urology, plastic surgery, paediatric surgery, or general surgery. Within the pelvic floor Master’s course, the other medical specialties with the highest participation, after gynecology (70.7%), were general surgeons (14.6%) and urologists (12.2%).

#### 3.1.2. Years of Expertise and Dedication

In all of the groups surveyed, the years of experience had a very wide range. In group 1, 76.6% had less than 10 years of experience. In group 2, 75% had less than 10 years. In group 3, 77% more than 11 years and in group 4, 56.1% had more than 11 years. (See more details in Table 1).

The vast majority of responders were not specifically dedicated in their daily practice to genitourinary malformations, including the group attending the European Society of Human Reproduction and Embryology (ESHRE) workshop (53.8%) (Table 1).

#### 3.1.3. Training in Malformations

Regarding specific training in anomalies, most gynecologists from both Valencian (55%) and Murcian (85%) scientific societies had not received specific training, and only a minor percentage (31.7% and 10%, respectively) had received exclusively theoretical training. Most gynecologists were not performing neovagina surgery. The respondents to the ESHRE workshop had mostly (53.8%) received theoretical and practical training. Within all of the groups, the most used technique, if any, was the McIndoe (38.5% in the ESHRE group, 15% in the Valencian gynecological Society (SOGCV) group, and 10% in the Murcian Gynecological Society (SGM) group), with the exception of the responders from the Master’s group, for which the Vecchietti procedure was the most used technique (4.76%) (Table 1).

#### 3.1.4. Responses after Viewing the Video

After viewing the video, a significant majority (92.7%–100%) from all interviewed groups recognized the potential utility of training using these feminized cadaver models (Table 1).

○Regarding the question of which technique seemed easier to perform, the most frequent answer within all groups was the modified McIndoe (70% for the SGM group, 53.8% for the ESHRE group, 46.7% for the SOGV group, and 39% for the PFM group) (Table 1).○When participants were questioned about the surgical technique they would prefer to train, the most frequent response within all groups was vulvo-perineal flaps (45% in the SOGV group, 38.5% in the ESHRE group, and 36.6% in the Master of pelvic floor dysfunctions (PFM) group) with the exception of the SGM gynecologists who opted mostly for the McIndoe technique (50%) (Table 1).○Regarding the reasons related to that decision, the most frequent answers were that they would choose the simplest technique, followed by the most efficient technique (Table 1).○Finally, a significant majority of participants (87.9%–100%) agreed that it should be mandatory to perform training on the cadaveric model before performing it on patients (Table 1).

## 4. Discussion

At present, there is no consensus in the literature regarding the optimal surgical technique to achieve the best functional outcome and sexual satisfaction [11]. Historically, the most common surgical procedure used to create a neovagina has been the modified Abbe–McIndoe operation. In our survey, this was the surgical technique most used within all surveyed groups. This procedure involves the dissection of a space between the rectum and bladder, placement of a stent covered with a split-thickness skin graft into the space, and the diligent use of vaginal dilation postoperatively. In the video, a modification of the McIndoe technique is presented. Such a modification avoids the use of a skin graft by using a polylactic acid vaginal prosthesis covered by Interceed^®^, which favors re-epithelialization of the surgical bed [12]. This fact is important because it simplifies the surgery, avoiding complications and maintaining its effectiveness. The simplification of the surgical technique was the most frequent reason regarding the criteria for choosing a technique to employ. It is possible that, for this reason, the majority responded that this was the simplest technique and, in the group of gynecologists with less training in the neovagina techniques (those from the Murcian Gynecological Society), the technique that they would prefer to learn. Within the other groups with a higher percentage of training and experience performing neovagina procedures, the technique that responders would like to train was vulvoperineal flaps because of its simplicity. It is possible that other reasons lie behind this decision, such as an individual wishing to learn a technique that they do not practice. However, it is also a very visual technique and, therefore, the video is highly informative regarding performance of this procedure. Other reasons could be that the vaginal route is the choice of many gynecologists, particularly those undertaking a Master’s of pelvic floor pathologies. Other procedures for the creation of a neovagina shown in the video include laparoscopic approaches. These are the laparoscopic Vecchietti [13] procedures, which are part of a modification of the open technique in which a neovagina is created using an external traction device affixed temporarily to the abdominal wall [14]. It also includes the Davydov technique, which is developed as a three-stage operation that requires dissection of the rectovesical space with abdominal mobilization of a segment of the peritoneum, and subsequent attachment of the peritoneum to the introitus [15]. Other vaginoplasty options (not included in the video) include the use of bowel, buccal mucosa, amnion, and various other allografts.

Our results confirm that the vast majority of specialist doctors were not specifically dedicated to genitourinary malformations, including those gynecologists from the group attending the ESHRE workshop with a special interest in adolescent pathologies. For this reason, in cases in which a surgical intervention is required, the patient should be referred to centers that have healthcare providers with expertise in this area. Alternatively, such an option should be at least considered because few surgeons have extensive experience in the construction of the neovagina, and surgery by a trained surgeon offers the best opportunity for a successful result [3]. Regardless of the surgical technique chosen, referrals to centers with expertise should be considered. These centers can offer the best counselling and management from a multidisciplinary point of view. The surgeon must be experienced with the procedure because the initial procedure is more likely to succeed than follow-up procedures [3]. The challenge, however, is that such centers, at least in some countries, are not clearly accredited, which complicates the correct referral of patients. Therefore, we believe that the accreditation of these reference centers in each country is necessary. In the opinion of the authors, to be able to access such accreditation, the appropriate education and adequate training is necessary. In university hospitals that wish to be accredited, access to cadaveric models to train these techniques before performing the procedure on patients should be the norm. This was indicated by the survey participants. Training by use of the cadaveric model and the volume of patients received would guarantee the correct training and expertise in this otherwise infrequent surgery. Human cadavers are an example of a high-fidelity simulator, clearly offering a more realistic anatomy and better tissue feel without the distraction of bleeding. Learners report a high degree of realism with this model, which closely resembles surgery in the real patient. Several studies [16] concluded that cadaveric skill courses focus on fundamental maneuvers with objective confirmation of success providing a viable adjunct to clinical operative experience. Costs associated with cadavers vary widely, but may limit their widespread use. The cost of a single cadaver in this study was over $1500, and this is the main limitation of the cadaveric model. However, although fresh cadavers are typically useful for 2 to 4 h, those fixed with the Thiel technique can be used for a number of days [17]. Thus, despite the previous assertion, in our experience, the results in terms of efficiency with the cadaveric model are highly satisfactory. The strengths of the study include the use of a cadaver model for rare surgeries, creation of a video, and creation of a survey tool. Regarding limitations, we recognize the bias in survey studies particularly due to the relatively low rate of responses obtained and the heterogeneous groups of interviewed specialists. For these reasons, we cannot generalize the obtained results. However, this study provides a starting point for future development.

Furthermore, the objective of this article was to carry out a “proof of concept” with this model. To validate its usefulness, a future study could conduct a baseline assessment of the knowledge regarding neovagina surgery, and follow-up the learning with a similar survey after the intervention (video education).

## 5. Conclusions

In our survey, the McIndoe operation was the surgical technique used predominantly by all groups. In the video, a modification of the McIndoe technique that avoids the use of skin grafts was presented, and this was chosen by a majority of respondents as the simplest surgical technique. A significant majority of specialist doctors were not specifically dedicated to genitourinary malformations. Nonetheless, they agreed that the use of feminized male cadavers was a useful approach to teaching different neovagina surgical techniques, and that the training of these techniques on cadavers should be suggested before procedures are carried out on a live patient.

## Figures and Tables

**Table 1 jcm-09-03722-t001:** Answers according to the different groups.

	Group 1 SOGCV ^1^ (*n* = 60)	Group 2 SGM ^2^ (*n* = 20)	Group 3 ESHRE ^3^ (*n* = 13)	Group 4 PFM ^4^ (*n* = 40)
1. speciality	100% gynecologist	100% gynecologist	92.3% gynecologist	70.7% gynecologist
2. years of expertise	43.3% > 20 33.3% between 11–20 15% between 5–10 8.4% > 5	50% > 20 25% between 11–20 15% between 5–10 10% > 5	46.2% > 20 30.8% between 11–20 15,4% between 5–10 7.6% > 5	24.4% > 20 31.6% between 11–20 22% between 5–10 22% > 5
3. dedication yes/no	95% not specifically	95% not specifically	53,8% not specifically	80.5% not specifically
4. training in malformations	55% none 31.7% theory 13.3% practical and theory	85% none 10% theory 5% practical and theory	53.8% practice and theory 23.1% theory 15.4% only practical 7.7% none	48.8% none 34.1% theory 12.2% practical and theory 4.7% only practical
5. most used technique	76.7% none 15% McIndoe 6.7% Vechietti 1.6% Davidoff	85% none 10% McIndoe 5% Vechietti	38.5% McIndoe 30.8% none 23.1% Davydov 7.6% Vechietti	87.8% None 7.2% Vecchietti 2.5% Mc Indoe 2.5% Flaps*
6. usefulness of feminized cadavers (agree/disagree))	93.4% agree	94% agree	100% agree	92.7% agree
7. easiest technique	46.7% McIndoe 26.7% Flaps 20% Vecchietti 6.6% Davidoff	70% McIndoe 25% Vecchietti 5% Flaps	53.8% McIndoe 23% Flaps 15.4% Vecchietti 7.8% Davidoff	39% McIndoe 36.6% Flaps 14.6% Davidoff 9.8% Vechietti
8. prefer to train	45% Flaps 25% Vecchietti 21.7% McIndoe 8.3% Davidoff	50% McIndoe 25% Vecchietti 15% Flaps 10% Davidoff	38.5% Flaps 30.8% McIndoe 23.1% Vecchietti 7.6% Davidoff	36.6% Flaps 29.3% McIndoe 24.4% Davydov 9.8% Vechietti
9. reasons to train	43.3% efficiency 35% easiness 21.7% safeness	45% easiness 35% safeness 20% efficiency	38.4% efficiency 30.8% easiness 30.8% safeness	41.5% easiness 34.1% efficiency 24.4% safeness
10. suggested training in cadaver (agree/disagree)	96.7% agree	95% agree	100% agree	87.9% agree

^1^ SOGCV: Members of Society of Obstetrics and Gynecology of the Valencian Community. ^2^ SGM: Members of Murcian Gynecological Society. ^3^ ESHRE: Members of the European Society of Human Reproduction and Embryology. ^4^ PFM: Members of the Pelvic Floor Master. Flaps* = Vulvoperineal Flaps.

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
