# Peer review of "Could Training in an Anatomical Model Be Useful to Teach Different Neovagina Surgical Techniques? A Descriptive Study about Knowledge and Experience of Techniques for Neovagina Surgery"

_jcm, 2020, doi:10.3390/jcm9113722_

Round 1

Reviewer 1 Report

In general this manuscript needs significant editing to improve language and grammar.

I don't think that the paper addresses the question.  The participants watched a video of a training technique. They didn't actually do the training, so they can't really say if it is useful or  not. The most the authors can conclude is that this idea was well accepted by participants. Otherwise it is a descriptive study about the knowledge and experience of a variety of surgeons with vaginal construction.

Author Response

Response to the reviewers of Manuscript jcm-976742

Dear Editor and Reviewers. Thank you very much for all the comments. We are sure that all of them will contribute to improve the manuscript

Reviewer 1

In general this manuscript needs significant editing to improve language and grammar.

I don't think that the paper addresses the question.  The participants watched a video of a training technique. They didn't actually do the training, so they can't really say if it is useful or  not. The most the authors can conclude is that this idea was well accepted by participants. Otherwise it is a descriptive study about the knowledge and experience of a variety of surgeons with vaginal construction.

Response: Thank you very much for your comments. We have sent the article for professional editing and the title has been modified following your suggestion. The new tittle is: Could training in an anatomical model be useful to teach different neovagina surgical techniques? A descriptive study about knowledge and experience of techniques for neovagina surgery

I attach below the certificate of the english edition . You can check in tne reviewed manuscript  all the changes that the authors have done  following your comments.

Reviewer 2 Report

Very interesting use of feminized male cadavers for this study.  Dissection of the neovagina for transgender surgery is based recognition of the prostate, and separating the prostate from the rectum.  How similar is this technique to the dissection for vaginal agenesis?  It seems that most surgeons going through this training program did not perform neovagina surgery.  

Author Response

Reviewer 2

Very interesting use of feminized male cadavers for this study.  Dissection of the neovagina for transgender surgery is based recognition of the prostate, and separating the prostate from the rectum.  How similar is this technique to the dissection for vaginal agenesis?

Response:

Thank you very much for your comments.

All the models were prepared by a trained urologist. And due to the number of the participants and models available, the neovagina surgery was a step to step technique performed by experts to small groups during the meeting. The comments from the experts that were not all familiarized with cadaveric surgery were that dissection of the spaces and feel during the practice of the neovagina surgery were similar to the reality and they even did not notice a difference when performing the surgery on a male cadaver.

A paragraph is added to section 2, experimental section (lines 191-195). The preparation of the feminized cadavers included removal of the penis together with the testicles and the reconstruction of the labia using the skin of the penis and scrotum. The space between the base of the scrotum and the anus was exposed to perform the different "neovagina" procedures. This preparation is a novel approach for the hands-on training of neovagina surgery on cadaveric models and allows a very real dissection of the spaces.

It seems that most surgeons going through this training program did not perform neovagina surgery.  

Response: Yes, in fact this training program was done in a course about adolescent pathologies organized by the ESHRE trying to address the need of developing skills before performing real life surgery. The majority of the assistants were gynecologist with special interest in these pathologies, but it is true that there were few specialist with great experience since this is a rare pathology. Although it would be ideal to send the survey to surgeons with experience in these techniques, there is not a relation of specialists dedicated to these problems across Europe.

Reviewer 3 Report

Re: Can the training the anatomical model be useful to learn different surgical techniques of Neovagina?

Review Report

This is a survey study describing the use of cadaveric models for teaching surgical techniques for creation of neovagina. The aim of the study is to evaluate the usefulness of a cadaveric model for teaching the surgical steps of four different neovagina creation techniques and to assess opinions regarding the use of this model. The authors created a video to illustrate the four different surgical techniques for neovagina with a cadaveric model and administered a survey to assess surgeon opinion of the model based on viewing the video. The authors concluded that surgeons found the cadaveric models useful for learning neovagina creation techniques and recommended use of these models in surgical training.

This a novel study regarding the use of cadaveric models for teaching neovagina surgical techniques. Given that neovagina creation procedures are infrequently performed, the authors describe a teaching tool to increase training in these procedures. The authors should be commended on the study of a relatively rare surgery. The strengths of the study include use of a cadaver model for rare surgeries, creation of a video, and creation of a survey tool.

While the relative lack of training on neovagina surgery is described, this study does not provide information as to whether the cadaveric model actually improves training for these surgical techniques and only provides surgeon opinions based on a short video. The use of cadaveric models for surgical training is already well supported; therefore, although rare surgical techniques are presented, the study does not add much information to show validation of the cadaveric model for these novel techniques. We therefore theoretically understand that cadaveric models would be helpful but do not know whether this translates to practice. The authors should expand on this further. More information should also be provided on the groups that were surveyed and how they were chosen as the generalizability and external validity of this study is difficult to interpret without those details.

It is therefore our recommendation that this manuscript be reconsidered after major revisions. Specific comments regarding the manuscript are below:

Specific comments to the author:

I commend the authors on the study of a cadaveric model for relatively rare surgical techniques. The study highlights the lack of training for neovaginal creation procedures among gynecologic and non-gynecologic surgeons. The authors illustrate that the use of cadaveric models to aid in training of these surgical techniques is viewed favorably among surgeons. My recommendations regarding specific areas of the manuscript are below:

Abstract

-Aims are clearly stated in the abstract

-Line 28: would replace “training” with “teaching”

-Line 29: would specify that the most employed technique refers to what surgeons said they used in practice, prior to watching video

-Line 33: “should be mandatory to perform” would re word; from this study, we can say that it is recommended but making the claim that is mandatory requires more data on actual use of this model for training

Introduction

-Line 37: would reword “most commonly cited prevalence” to “prevalence”

-Line 39: is androgen insensitivity syndrome the second most common cause of vaginal agenesis? Would state that more explicitly as this sentence does not fit into flow of the text as it is currently written

-Would make description of different surgical models here more concise

-Line 60: would replace “corpses” with “cadavers”

-Line 62: although there are more ethical issues with use of animals, there are also different ethics involved with use of cadavers; would rephrase as decreased ethical risks instead of “avoiding ethical concerns”

-Line 65-66: would remove “in comparison with other learning models” as other models were not compared

Experimental Section

-Line 73: would replace “lips” with “labia”

-Line 110 and on: replace “malformation” with “anomaly”

-Line 102: unclear what “pelvic floor master” means

-No information is provided about how the questionnaire was created. Was it based off of previous, validated questionnaires used in medical education?

-Recommend including more information here about the different surgical techniques. Although some of this is described in discussion, the description of techniques should be added to the methods. As the description of the techniques in the video is quite fast, it may be useful to include an appendix with descriptions of the different surgical techniques.

-Do we know if the audience is familiar with the use of Thiel embalmed cadaver models? Would add more on that here as video is quick and hard to understand if unfamiliar with anatomy of the models. Were the viewers in this study given more information on anatomy of these models before watching video?

-I would also recommend providing more information on why the four groups that were surveyed were chosen here. The average reader will not necessarily know the composition of these groups and whether performing neovagina procedures would be considered in the scope of their practice.

-Multiple links are provided for survey but since survey is the same it can only be linked once in the document

-Two of the questions in the survey say “totally disagree,” “agree,” and “totally disagree.” Please provide an explanation for why only these three options were chosen, as in why was “disagree” and “neither agree nor disagree” not included as this can affect interpretation of results

-Discuss whether this study was approved or exempt by an institutional review board

Results

-What was the survey response rate? How many students/learners/trainees participated in the course?

-For the results section, I would recommend using Table 2 to support the reported data. The descriptive numerical data from Table 2 would make the reporting of results more robust than using descriptive terms only (such as majority, minority etc.).

-Recommend reformatting results from a bulleted format to paragraph form.

-No comparative statistics were performed, only descriptive. As the authors stated in their introduction, the objective of the study was to determine the usefulness of the training videos they developed; however, how the authors chose to define “usefulness” is not clearly stated. Thus, it is difficult to interpret the survey results.  Perhaps, performing a baseline assessment of knowledge regarding neovaginas and then following up with a similar survey after the intervention (video education) would have been a better way to determine the “usefulness” of the activity.

-Provide more information on how many surveys were sent vs who answered the survey.

-Line 121: change “visualizing” to “viewing”

Discussion

-Lines 150-151: would move this to introduction and instead focus discussion on only the use of surgical techniques

-Good discussion here to highlight why proper training and proper patient referral is needed for these procedures

-Line 204: would reword “learners feel they are really operating” to a more formal statement

-Would include a discussion here of future direction and specifically how to test the validity of this cadaveric model in practice; how is this information from this study going to be used to inform training and clinical practice of these surgeries?

-Would comment on bias in survey studies

-Would also comment here on the generalizability of these study findings

Conclusions

-Line 218: change “corpse” to “cadaver”

Tables

-Table 1 is the survey; this could instead be included as an appendix. If it is reported as a table, clear table headings are needed.

Author Response

Reviewer 3

This is a survey study describing the use of cadaveric models for teaching surgical techniques for creation of neovagina. The aim of the study is to evaluate the usefulness of a cadaveric model for teaching the surgical steps of four different neovagina creation techniques and to assess opinions regarding the use of this model. The authors created a video to illustrate the four different surgical techniques for neovagina with a cadaveric model and administered a survey to assess surgeon opinion of the model based on viewing the video. The authors concluded that surgeons found the cadaveric models useful for learning neovagina creation techniques and recommended use of these models in surgical training. 

This a novel study regarding the use of cadaveric models for teaching neovagina surgical techniques. Given that neovagina creation procedures are infrequently performed, the authors describe a teaching tool to increase training in these procedures. The authors should be commended on the study of a relatively rare surgery. The strengths of the study include use of a cadaver model for rare surgeries, creation of a video, and creation of a survey tool. 

While the relative lack of training on neovagina surgery is described, this study does not provide information as to whether the cadaveric model actually improves training for these surgical techniques and only provides surgeon opinions based on a short video. The use of cadaveric models for surgical training is already well supported; therefore, although rare surgical techniques are presented, the study does not add much information to show validation of the cadaveric model for these novel techniques. We therefore theoretically understand that cadaveric models would be helpful but do not know whether this translates to practice. The authors should expand on this further. More information should also be provided on the groups that were surveyed and how they were chosen as the generalizability and external validity of this study is difficult to interpret without those details.

 Response: We agree with the reviewer and thank you for this comment. The design of this study was not correct to validate the proposed model. The exact aim is to show a new cadaveric model for this rare surgery. The way to show it was to record a video presenting four different techniques to perform neovagina. In this sense this article could be considered as a “proof of concept” to show this model but not to validate its application. We also agree with the idea of the reviewer to expand this article and create a validation of these model. Therefore we have modified the article, especially the abstract and aim in this sense.

It is therefore our recommendation that this manuscript be reconsidered after major revisions. Specific comments regarding the manuscript are below: 

Specific comments to the author: 

I commend the authors on the study of a cadaveric model for relatively rare surgical techniques. The study highlights the lack of training for neovaginal creation procedures among gynecologic and non-gynecologic surgeons. The authors illustrate that the use of cadaveric models to aid in training of these surgical techniques is viewed favorably among surgeons. My recommendations regarding specific areas of the manuscript are below: 

Abstract

-Aims are clearly stated in the abstract

Response : We have modified the aim be more accurate. (lines 28-30)

-Line 28: would replace “training” with “teaching”

Response: We have already made this change at line 30

-Line 29: would specify that the most employed technique refers to what surgeons said they used in practice, prior to watching video

Response: We have already made this change at line 36 : “Before watching this video, the most employed technique were McIndoe and Vecchietti procedures”

-Line 33: “should be mandatory to perform” would re word; from this study, we can say that it is recommended but making the claim that is mandatory requires more data on actual use of this model for training.

Response: We have already made this change at lines 39-41: ”Use of this model was considered useful for training in neovagina surgical techniques and moreover, that it should be recommended before performing in the real patient. Further investigation is needed to validate this model”.

Introduction

-Line 37: would reword “most commonly cited prevalence” to “prevalence”

Response: We have already made this change at line 147: “The prevalence for vaginal agenesis due to Rokitanky Sindrome is 1 in 5000 (range 1 per 4000 to 10,000 females)”

-Line 39: is androgen insensitivity syndrome the second most common cause of vaginal agenesis? Would state that more explicitly as this sentence does not fit into flow of the text as it is currently written

Response: We have made this change at lines 148-150: ” The second pathology responsible for vaginal agenesis and the need of neovagina surgery is androgen insensitivity syndrome whose incidence in females is 1:20,000 [2]

-Would make description of different surgical models here more concise

Response: We have added a brief description of these techniques at lines 161-163. Besides, we have a new Appendix A where these techniques has been described

-Line 60: would replace “corpses” with “cadavers”

Response: We have already made this change at this line (now 179) and along the text.

-Line 62: although there are more ethical issues with use of animals, there are also different ethics involved with use of cadavers; would rephrase as decreased ethical risks instead of “avoiding ethical concerns”

Response: We have modified this sentence at lines 179-181: ”Use of human cadavers is a simulation aid that allows for surgical practice via extremely detailed and lifelike reproductions of each structure and decreased ethical risk associated with the use of animal surgical laboratories

-Line 65-66: would remove “in comparison with other learning models” as other models were not compared

 Response: We have modified this sentence at line 65: “in comparison with other learning models” has been deleted.

Experimental Section

-Line 73: would replace “lips” with “labia”

 Response: We have modified this sentence at line 426.

-Line 110 and on: replace “malformation” with “anomaly”

Response: We have modified this sentence at line 135.

-Line 102: unclear what “pelvic floor master” means

Response: We have added information about the groups at lines 235-241: Pelvic floor master means “This University Master on multidisciplinary Pelvic Floor Surgery provides a multidisciplinary professional update on pelvic floor diseases and on their management. The Master course includes an update on anatomy, on pathophysiology and on clinical workup of the patient with pelvic floor dysfunction, including urogenital anomalies. In particular, the course provides hands-on teaching on the management of these patients and on the multidisciplinary instrumental in the dissection room using Thiel embalmed cadavers”

-No information is provided about how the questionnaire was created. Was it based of previous, validated questionnaires used in medical education?

Response: Since we did not find validated questionnaires used in medical education for this issue, prior to running the survey, we did a short pilot test with 5 students to examine comprehension of the questions. After that the survey was sent to the rest of participants (lines 219-224).

-Recommend including more information here about the different surgical techniques. Although some of this is described in discussion, the description of techniques should be added to the methods. As the description of the techniques in the video is quite fast, it may be useful to include an appendix with descriptions of the different surgical techniques.

Response: As you suggested, we have already added some information in the introduction section (lines 161-163). Besides, we agree with the idea of including an appendix with the description of these techniques and we have added this at line 212-213: “We performed recordings of four different surgical procedures to perform neovagina (Modified McIndoe, Modified Vecchietti, Davydov and Vulvoperineal pediculated Flaps) (see Appendix A at lines 592-623)”

-Do we know if the audience is familiar with the use of Thiel embalmed cadaver models? Would add more on that here as video is quick and hard to understand if unfamiliar with anatomy of the models. Were the viewers in this study given more information on anatomy of these models before watching video?

Response: Yes, in the survey we explain that Thiel embalmed is a method which allows to move the cadavers to gynecological exploration and allows to perform laparoscopic procedures in a very similar manner as it would be done with patients. Besides, the participants in the ESHRE course and in the pelvic floor master interacted with the cadavers in the dissection room. Additionally, in the presentation of the survey we explained that we used “feminized” male cadavers to perform neovagina surgery. The preparation of the feminized cadavers included removal of the penis together with the testicles and the reconstruction of the labia using the skin of the penis and scrotum. The space between the base of the scrotum and the anus was exposed to perform the different "neovagina" procedures. This preparation is a novel approach for the hands-on training of neovagina surgery on cadaveric models and allows a very real dissection of the spaces since the rest of the anatomy remains intact. We have added these explanations also in the article at lines 224-227. “In the presentation of the survey we explained that we used “feminized” male cadavers Thiel embalmed to really perform neovaginas. We clarified that this embalmed method allows to move the cadaver to do vaginal surgery and also laparoscopic procedure in a very similar manner as it would be done with patients.”

-I would also recommend providing more information on why the four groups that were surveyed were chosen here. The average reader will not necessarily know the composition of these groups and whether performing neovagina procedures would be considered in the scope of their practice.

Response: We have completed this information at lines 230-245. “The survey was sent to four different groups: 1) Attendants and speakers to the ESHRE (European Society of Human Reproduction and Embryology) Campus Symposium “Gynecological pathologies at adolescence” organized at Miguel Hernandez University, San Juan, Alicante, Spain. These participants were chosen since they watched this hands-on session live when it was broadcasted to the audience, 2) Students and professors of the “Master of pelvic floor dysfunctions” of Miguel Hernandez University, Elche, Alicante, Spain. This University Master on multidisciplinary Pelvic Floor Surgery provides a professional update on pelvic floor diseases and on their management. The Master course includes an update on anatomy, on pathophysiology and on clinical workup of the patient with pelvic floor dysfunction, including urogenital anomalies. In particular, the course provides hands-on teaching on the management of these patients and on the multidisciplinary instrumental in the dissection room using Thiel embalmed cadavers, 3) Gynecologists affiliated to the Valencian gynecological Society (SOGCV) and 4) Murcian gynecological Society (SGM). These participants were chosen to explore the opinion of the general gynecologist about the method while the other groups would be considered to have some experience with Thiel embalmed cadavers and to have neovagina procedures in the scope of their practice.

-Multiple links are provided for survey but since survey is the same it can only be linked once in the document

 Response: We agree with you. We have kept only one link to the survey. The others have been deleted.

-Two of the questions in the survey say “totally disagree,” “agree,” and “totally disagree.” Please provide an explanation for why only these three options were chosen, as in why was “disagree” and “neither agree nor disagree” not included as this can affect interpretation of results.

Response: We use these 3 parameters commonly used on Likert scales “totally agree,” “agree,” and “totally disagree. The option proposed by the reviewer could also be valid, but in this case it does not affect the interpretation of the results obtained as the vast majority of responses belong to the same category.

-Discuss whether this study was approved or exempt by an institutional review board

Response: The ESHRE (European Society of Human Reproduction and Embryology) requested permission from Miguel Hernandez University to carry out this course in the dissection room and to be able to record the interventions, and it was authorized. Subsequently, the authors requested ESHRE, SOGV, SGM and the Coordinator of Pelvic Floor Master allowance and support to disseminate the educational video among those attending the course and theirs members as well as to use the results for this article, and it was also authorized.

Results

-What was the survey response rate? How many students/learners/trainees participated in the course?

The rate of answers was 25,9% (it has been added at line 249).  There were 63 participants in the course including 5 trainees, 7 teachers and 51 students.

-For the results section, I would recommend using Table 2 to support the reported data. The descriptive numerical data from Table 2 would make the reporting of results more robust than using descriptive terms only (such as majority, minority etc.).

Response: We have completed the result sections with the numbers included in the table.

-Recommend reformatting results from a bulleted format to paragraph form.

Response: This is a specific request of the template for this journal. If the reviewer and editor agree I can modify the format.

-No comparative statistics were performed, only descriptive. As the authors stated in their introduction, the objective of the study was to determine the usefulness of the training videos they developed; however, how the authors chose to define “usefulness” is not clearly stated. Thus, it is difficult to interpret the survey results.  Perhaps, performing a baseline assessment of knowledge regarding neovaginas and then following up with a similar survey after the intervention (video education) would have been a better way to determine the “usefulness” of the activity.

Response: We have already change all the manuscript in this sense. The design of this study was not correct to validate the proposed model. The exact aim is to show a new cadaveric model for this rare surgery. The way to show it was to record a video which allowed to show four different techniques to perform neovagina. In this sense this article could be considered as a “proof of concept” to show this model but not to validate its application. We also agree with the idea of this reviewer to expand this article and create a validation of this model. We have modified the article, especially the abstract and aim in these way.

-Provide more information on how many surveys were sent vs who answered the survey.

Response: We sent 512 surveys and we got 133 responses.

-Line 121: change “visualizing” to “viewing”

Response: We have changed it (lines 435 and 437)

 Discussion

-Lines 150-151: would move this to introduction and instead focus discussion on only the use of surgical techniques

Response: We have moved the first paragraph of the discussion to the introduction, as the reviewer suggested.

-Good discussion here to highlight why proper training and proper patient referral is needed for these procedures

Response: Thank you very much for this comment.

-Line 204: would reword “learners feel they are really operating” to a more formal statement

Response: We have removed this sentence and we have added the following:” Learners report a high degree of realism with this model, which closely resembles surgery in the real patient” (lines 543-544)

-Would include a discussion here of future direction and specifically how to test the validity of this cadaveric model in practice; how is this information from this study going to be used to inform training and clinical practice of these surgeries?

Response: We have added this paragraph: “However, the objective of this article was to carry out a "proof of concept" with this model. To validate its usefulness, another future study could be to study a baseline assessment of knowledge regarding neovagina surgery and then follow up the learning with a similar survey after the intervention (video education)” at lines 556-559

-Would comment on bias in survey studies-Would also comment here on the generalizability of these study findings

Response:  We have added this paragraph:”The strengths of the study include use of a cadaver model for rare surgeries, creation of a video, and creation of a survey tool. Between the limitations we recognized the bias in survey studies especially due to the relatively low rate of responses obtained and having interviewed heterogeneous groups of specialists. For these reasons, we cannot generalize the results obtained in this study but it gives us a start point for future development” at lines 551-555.

 Conclusions

-Line 218: change “corpse” to “cadaver”

 Response: We have already changed it as it has been changed all along the text.

Tables

-Table 1 is the survey; this could instead be included as an appendix. If it is reported as a table, clear table headings are needed.

Response: The table 1 is now appendix B

Round 2

Reviewer 3 Report

I would recommend that the authors edit the survey responses to include all the survey results. The way in which the data are presented is not clear and incomplete. For example, in group 3, only 92% were gynecologists, but what specialty was the remaining 8%?

Otherwise, the edits are sufficient. 

Author Response

Response to the reviewer 3 of Manuscript jcm-976742

Thank you very much for your comments. We are sure that all of them will contribute to improve the manuscript.

Reviewer 3: I would recommend that the authors edit the survey responses to include all the survey results. The way in which the data are presented is not clear and incomplete. For example, in group 3, only 92% were gynecologists, but what specialty was the remaining 8%?

Response: We have already re-edited Table 1 including all the survey results. You can check the new changes highlighted in yellow in the manuscript.

Group 1

SOGCV1

(N = 60)

Group 2

SGM2

(N = 20)

Group 3

ESHRE3

(N = 13)

Group 4

PFM4

(N = 40)

 1. Specialty

100% gynecologist

100% gynecologist

92,3% gynecologist

70,7% gynecologist

2. Years of expertise

43.3%> 20

33.3%between11-20

15% between 5-10

8.4% > 5

50%> 20

25%between11-20

15% between 5-10

10% > 5

46.2%> 20

30.8%between11-20

15,4%between 5-10

7.6% > 5

24.4%> 20

31.6% between 11-20

 22% between 5-10

22% > 5

3. Dedication yes/no

95% not specifically

95% not specifically

53,8% not specifically

80.5% not specifically

4. Training in malformations

55% none

31.7% theory

13.3% practical and theory

85% none

10% theory

5% practical and theory

53.8% practice and theory

23.1% theory

15.4% only practical

7.7% none

48.8% none

34.1% theory

12.2%practical and theory

4.7% only practical

5. Most used technique

76.7% none

15% McIndoe

6.7% Vechietti

1.6% Davidoff

85% none

10% McIndoe

5% Vechietti

38.5% McIndoe

30.8% none

23.1% Davydov

7.6% Vechietti

87.8% None

7.2%Vecchietti

2.5% Mc Indoe

2.5% Flaps*

6. Usefulness of feminized cadavers (agree/disagree))

93.4% agree

94% agree

100% agree

92.7% agree

7. Easiest technique

46.7% McIndoe

26.7% Flaps

20% Vecchietti

6.6% Davidoff

70% McIndoe

25% Vecchietti

5% Flaps

53.8% McIndoe

23% Flaps

15.4% Vecchietti

7,8% Davidoff

39% McIndoe

36.6% Flaps

14.6% Davidoff

9.8% Vechietti

8. Prefer to train

45% Flaps

25% Vecchietti

21.7% McIndoe

8.3% Davidoff

50% McIndoe

25% Vecchietti

15% Flaps

10% Davidoff

38.5% Flaps

30.8% McIndoe

23.1% Vecchietti

7.6% Davidoff

36.6% Flaps

29.3% McIndoe

24.4% Davydov

9.8% Vechietti

9. Reasons to train

43.3% efficiency

35% easiness

21.7% safeness

45% easiness

35% safeness

20% efficiency

38.4% efficiency

30.8% easiness

30.8% safeness

41.5% easiness

34.1% efficiency

24.4% safeness

10. Suggested training in cadaver (agree/disagree)

96.7% agree

95% agree

100% agree

87.9% agree

We have also added in the results section  this phrase at lines 159-161:” In the ESHRE group, 12/13 (92.3%) were gynecologist. Only 1/13 (7.7%) had other speciality different from gynecology, urology, plastic surgeon, pediatric surgeon or general surgeon”(159-161)

And we have also modified the paragraph of “years of expertise and Dedication”. We have added at lines  167-169: “In group 1, 76.6% had less than 10 years of experience. In group 2, 75% had less than 10 years. In group 3, 77% more than 11 years and in group 4, 56.1% had more than 11 years. (see more details in Table 1).”

Finally, we have added “Table 1” at lines 172, 183,189,194,200,204 and 208. You can check the new changes highlighted in yellow in the manuscript.